# Phototherapy as a Treatment for Dermatological Diseases, Cancer, Aesthetic Dermatologic Conditions and Allergenic Rhinitis in Adult and Paediatric Medicine

**DOI:** 10.3390/life13010196

**Published:** 2023-01-09

**Authors:** Roy Kennedy

**Affiliations:** Warwickshire College University Centre, Warwick New Road, Royal Leamington Spa, Warwickshire CV32 5JE, UK; rkennedy@warwickshire.ac.uk

**Keywords:** phototherapy, skin conditions, allergy, rhinitis, blue light

## Abstract

The development of light-emitting diodes (LEDs) has led to an increase in the use of lighting regimes within medicine particularly as a treatment for dermatological conditions. New devices have demonstrated significant results for the treatment of medical conditions, including mild-to-moderate acne vulgaris, wound healing, psoriasis, squamous cell carcinoma in situ (Bowen’s disease), basal cell carcinoma, actinic keratosis, and cosmetic applications. The three wavelengths of light that have demonstrated several therapeutic applications are blue (415 nm), red (633 nm), and near-infrared (830 nm). This review shows their potential for treating dermatological conditions. Phototherapy has also been shown to be an effective treatment for allergenic rhinitis in children and adults. In a double-anonymized randomized study it was found that there was 70% improvement of clinical symptoms of allergic rhinitis after intranasal illumination by low-energy narrow-band phototherapy at a wavelength of 660 nm three times a day for 14 consecutive days. Improvement of oedema in many patients with an age range of 7–17 were also observed. These light treatments can now be self-administered by sufferers using devices such as the Allergy Reliever phototherapy device. The device emits visible light (mUV/VIS) and infra-red light (660 nm and 940 nm) wavelengths directly on to the skin in the nasal cavity for a 3 min period. Several phototherapy devices emitting a range of wavelengths have recently become available for use and which give good outcomes for some dermatological conditions.

## 1. Introduction

Non-invasive phototherapy procedures for medical and aesthetic dermatologic conditions are becoming more common. Phototherapy as a medical procedure is the use of non-thermal, non-invasive light for therapeutic medical applications. Usually, the therapy is provided by a variety of light-emitting devices which are based on light-emitting diode (LEDs) technology. Interest in recent advances in the use of LEDs has led to their application for a variety of medical and cosmetic uses [1]. The three wavelengths of light that have demonstrated several therapeutic applications are blue (415 nm), red (633 nm), and near-infrared (830 nm) [1]. Clinical studies demonstrated LEDs had beneficial effects on wound healing [2,3]. Clinical use of photobiomodulation (PBM) therapy, promotes surgical wound closure [4]. PBM treatments have specifically shown clinical benefits in the management of serious burns. Various PBM parameters have been examined in these studies including wavelengths (ranging from 660 nm to near-infrared 904 nm), pulsing (0–80 Hz) and doses (2–25 J/cm^2^). Several reviews have published evidence on the effectiveness of PBM treatments in mitigating inflammation and promoting wound healing [2]. Recent studies with blue light (470 nm) showed significant improvement in release of nitric oxide from nitrosyl complexes. Nitric oxide released through the inducible isoform (iNOS) is synthesized in the early phase of wound healing by inflammatory cells, mainly macrophages. It has been shown to regulate collagen formation, cell proliferation and wound contraction in animal models [5]. Patients affected by chronic wounds were treated with a blue LED Light medical device (EmoLED). Approximately 84% had a positive response to the treatment during a 10-week observation period [6].

This developing area of clinical phototherapy shows a promise in future therapeutic applications because the patient has the options in some cases to self-administer the application of the therapy. Further research is required to fully understand the beneficial effects of light treatment and determine any problematic responses.

## 2. Use of Phototherapy including Blue Light as a Dermatological Treatment and for Other Medical Conditions

### 2.1. Psoriasis (Chronic Autoimmune Condition)

Narrow Band ultraviolet B (NBUVB) (wavelength 311–313 nm) has been most studied and is frequently recommended as an effective treatment for paediatric patients with psoriasis. It has good efficacy, is safe and relatively easy to administer. In studies on psoriasis with 88 paediatric patients (average age of 12 ± 4 years) NBUVB therapy applied for 3.1 ± 2.26 months (average cumulative dose of 46.5 J/cm^2^) gave a 75% improvement in patients with full clearance achieved in 51% of cases [7]. Other effective wavelengths include broadband UVB (Broad Band ultraviolet B (BBUV), 290–320 nm) and ultra violet A UVA (320–400 nm) [8,9]. Thirty patients with psoriasis (mean age 11 ± 3.6 years) treated with BBUVB (mean treatment number 28.8 ± 13.3), showed that 93.3% of subjects had more than a 75% improvement in their condition. Phototherapy can also be combined with adjuvant therapies but there is some variability in the results. Topical agents had efficacy when combined with phototherapy. These included emollients (i.e., mineral oil), topical corticosteroids, vitamin D (applied after phototherapy), coal tar, and retinoids [10,11,12]. Phototherapy should be considered for older patients, especially if they have contraindications to other treatments. There is no consensus on the use of phototherapy on paediatric psoriasis. Other light wavelengths have also been shown to have beneficial effects on psoriasis. A study involving 30 adults with a mild form of *psoriasis vulgaris* not previously receiving treatment was carried out to investigate the effect of a device emitting blue LED light for 3 months. The treatment using blue LED light was shown to be a safe and highly effective way to treat psoriasis. Blue light normalizes the proliferation of keratinocytes, endothelial cells and fibroblasts which can promote skin healing without a cytotoxic effect [13,14].

### 2.2. Atopic Dermatitis (Eczema)

Phototherapy is considered as an effective therapy in paediatric patients with moderate-to-severe atopic dermatitis (AD). Both UVA and UVB are safe and effective treatments for children with AD [15]. Phototherapy is effective after the application of topical therapies and can be administered as an additional treatment if necessary. It can be used as a potential adjunctive treatment in older children. It is effective in the treatment of chronic, lichenified disease [16] although the low numbers of cases means it has not been well characterised. The preferred wavelength for treatment of AD in the paediatric populations is NBUVB. Trials with 905 patients having a mean age of 32 years (range 8–83 years), concluded NBUVB and medium dose UVA were the safest and most effective phototherapy treatments for AD patients [17]. The efficacy of irradiation with blue light (415 nm and 450 nm) was compared with placebo irradiation in adult patients with atopic dermatitis. It was demonstrated that blue light induces an anti-inflammatory and antiproliferative effect. Blue light may be beneficial for chronic inflammatory skin diseases such as atopic dermatitis, eczema, and psoriasis although further research would be necessary to confirm these full beneficial effects [18].

### 2.3. Pityriasis Lichenoides (Cutaneous Rash)

*Pityriasis lichenoides* (PL) causes an inflammatory skin condition which is difficult to treat. Evidence supporting the efficacy of phototherapy in paediatric patients with *Pityriasis lichenoides et varioliformis acuta* (PLEVA) and *Pityriasis lichenoides chronica* (PLC) is limited because of the small number of cases [19]. Five patients with PL treated with NBUVB showed complete success after treatment with 21 sessions of phototherapy [19]. The average cumulative phototherapy dose was 21 J/cm^2^. Studies on the effect of blue light (415 nm and 450 nm) on *Pityriasis Lichenoides* have not been reported but it is likely that it may be an effective treatment for this skin complaint as it shares some of the pathways noted in studies for other dermatological conditions [19]. Further studies would be required to confirm the effectiveness of blue light for this skin condition as there are small numbers of sufferers. The maximum penetration of blue light is 0.07–1 mm where it may interact with many chromophores including cytochromes which is advantageous.

### 2.4. Cutaneous T-Cell Lymphoma/Vitiligo (Cancer)

Both PUVA and NBUVB has been shown to be as effective treatments for paediatric cutaneous T-cell lymphoma (CTCL). Narrow Band ultra violet B is easier to administer and for this reason can be considered as the preferred treatment [20]. NBUVB phototherapy, combining ultraviolet A1 (UVA1) and UVB, 308 nm phototherapy and topical PUVA have been used effectively in the treatment of childhood vitiligo (loss of skin pigment cells) and pruritus (unpleasant skin sensation) [20]. Phototherapy has also been used successfully to treat scleroderma, morphea, nodular prurigo, Langerhans cell histiocytosis, and cutaneous graft-versus-host disease (GVHD) in paediatric patients [21]. Efficacy evaluation in these conditions was only considered for a few cases [20,21,22]. Vitiligo development is considered to be an autoimmune response with CD8+ cytotoxic T-lymphocytes inducing melanocyte apoptosis. Infiltrating T-cells in vitiligo are associated with a Th1 phenotype, which induces melanocyte destruction via direct cytotoxicity and modulation of cytokine microenvironment [23]. There is evidence that blue light increases in the number of T cells infiltrating the skin a factor which suggests blue light maybe an effective treatment for *Vitiligo* although studies would be required to show more evidence [24].

### 2.5. Use of Near Infra-Red (NIR) Wavelengths in Other Phototherapy Treatment Usage

NIR phototherapy has been used successfully to treat sleep bruxism in children [25]. Bruxism is a muscle activity which occurs during sleep. It is usually masticatory and is characterized as either rhythmic (phasic) or non-rhythmic (tonic) in otherwise healthy children [26]. Clenching/grinding the teeth while sleeping is a sleep bruxism, whereas clenching/grinding teeth is a secondary or awake bruxism if it occurs during the day.

A marked improvement of in oedema was also seen in these trials. This type of phototherapy can be a useful additional approach in the treatment of allergic rhinitis. Treated patients suffering from allergenic rhinitis were followed for 1 year and showed consistent improvement and a reduction in IgE antibodies (3 months after therapy) [27]. Clinical trials [27] using children aged 7–17 years old showed that treatment with visible (400–800 nm) and infrared (800–1000 m) wavelengths had a therapeutic benefit for allergenic rhinitis [28,29]. The two types of illumination have differences in their photochemical and photo physical properties. Visible light may stimulate metabolic events at the level of the respiratory chain of mitochondria, including the formation of reactive oxygen species. However, infrared illumination activates enzymes and probably also Ca+ channels in the membranes [26]. In vitro studies using infra-/red light have not shown any useful impact on wound healing by activating cells located in deeper skin layers such as fibroblasts or stem cells [2].

### 2.6. Phototherapy Using Blue Light Wavelengths for Treatment of Other Clinical Conditions

Clinical trials using blue light indicate promising results for the development of a system for improved wound healing through light stimulation. Blue light can be used to prevent non optimal epidermisation in premature healing stages by slowing down the cell metabolism [30]. An effect on keratinocytes, could be achieved using longer blue light irradiations with a maximum effect at 30 min [30]. In vitro studies treating different bacterial strains with blue light revealed bacteriostatic and even bactericidal effects. Blue light significantly decreases wound size seven days after the start of treatment, correlating with enhanced epithelialization [30]. The wound healing process is enhanced using blue light irradiation as it inhibits the formation of bacterial colonies and promotes optimal epidermisation by preventing premature wound closure.

Jaundice is commonly seen in patients with liver and gallbladder dysfunctions and its occurrence is closely related to the bilirubin functional metabolism in neonatal patients. Neonatal blue light phototherapy (NBLP) is an effective treatment for hyperbilirubinemia. However, a significantly higher prevalence of hyperpigmented skin patches was found in children treated with NBLP than in children without NBLP [31]. Further research would be necessary to fully understand the effectiveness of NBLP treatment.

Bright light therapy is also an effective treatment for seasonal affective depression (SAD). Blue light was equally effective as bright light in treating depression symptoms [32]. A study with 24 patients with SAD, showed that daily morning blue light therapy for three weeks was more effective than red light for improving depression [33]. Blue light was effective in reducing fatigue and daytime sleepiness in a trial with 30 people who had suffered a traumatic brain injury if used daily for 4 weeks [34]. Additional studies would be required to confirm these results.

Blue light has anti-microbial and anti-inflammatory effects without damaging the tissue in comparison to UV light [35]. Microbes can cause secondary damage on skin and therefore blue light could be a useful treatment for skin problems in children. Acne is a common paediatric condition caused by *Propionibacterium acnes*. Inflammation of the skin occurs when fat production increases as a result of an elevated hormone level. *Propionibacterium acnes* can be trapped in the sebaceous duct, the skin swells causing red spots, blisters, and eventually nodules and cysts. *P. acnes* can be killed by blue light. Application of blue light to the skin of patients with mild to moderate acne may also improve the wounds associated with longer term infection [36].

### 2.7. Phototherapy as a Treatment for Allergenic Rhinitis and Other Skin Conditions

Intranasal phototherapy using a combination of visible light and ultraviolet light (UVB, UVA) was effective in treating allergic rhinitis. The treatment reduced the number of inflammatory cells and level of mediators associated with this condition [36]. Phototherapy has also been shown to be an effective treatment for allergenic rhinitis in children. In a double-anonymized randomized study it was found that there was a 70% improvement of clinical symptoms of allergic rhinitis after intranasal illumination by low-energy narrow-band phototherapy at a wavelength of 660 nm when applied three times a day for 14 consecutive days [37]. Improvement of oedema in many patients with an age range of 7–17 were also observed [20]. Light-emitting diodes were also shown to have therapeutic potential. The outputs from LEDs can range from blue to violet (400 nm) to red (about 700 nm) but some LEDs emit infrared (IR) energy (wavelength 830 nm or longer). Phototherapy is a well-documented and effective treatment for a range of skin diseases. It can be used safely and effectively for skin diseases in children; however, consideration is needed for the use of this therapy in very young children. Phototherapy can reduce severe eczema, psoriasis, and a range of other skin problems with sustained remission after 12 months [16]. It is best used as second-line therapy after standard topical regimens have failed in children [12].

### 2.8. Photodynamic Therapy (Phototherapy Combined with 5-Aminolevulinic Acid)

Photodynamic therapy (PDT) uses a photosensitizer (light sensitive substance) in combination with exposure to the light wavelength corresponding to the absorbance of the sensitizer. Treatment with red and blue wavelengths including pulsed light are used in the treatment to activate the photosensitizer. The treatment works because of the effect of free radicals, which form in the presence of oxygen causing cell death [38]. Photodynamic therapy with 5-aminolevulinic acid (ALA) is the most common sensitizer used to treat conditions. This includes pre-cancerous and cancerous lesions [39,40] and photo aged skin [41,42]. PDT is also a suitable treatment for viral warts in children [43]. The efficacy of PDT for the treatment of Squamous cell carcinoma in situ (Bowen’s disease) using PDT and ALA as the photosensitizer has been reported [44]. ALA was applied topically four hours prior to PDT irradiation and resulted in clearance of all lesions after two treatments, which was a significant effect compared to a non PDT control treatment. No adverse events were associated with the use of PDT [43]. The relative efficacy of red and green light for the treatment of Bowen’s disease has been studied [45]. Initial clearance rate with red light was 94 percent versus 72 percent for green light. Over the following 12 months, there were two recurrences in the red-light group and seven in the green light group, reducing the overall clearance rates to 88 percent and 48 percent, respectively. Further research on the effect of red and green light should confirm these trial results. Use of PDT with ALA for basal cell carcinoma of the skin (BCC) was less effective. Relapse rates of 50 percent or greater have been reported [46]. However, applying a second PDT treatment after seven days (treatment with 630 ± 15 nm red light (120–134 J/cm^2^, 50 ± 100 mW/cm^2^) for one hour following application of 20%ALA) improved outcomes. A complete response rate of 100 percent was observed one month after treatment and the cosmetic results were observed to be good. Actinic keratosis (AK) is a pre-cancerous skin condition, which if left untreated may eventually progress to squamous cell carcinoma. It occurs as rough, scaly lesions following long-term sun exposure on the face and forearms of fair-skinned humans. It has been reported that PDT provides good treatment for this condition [47]. When PDT was used with 20% 5-aminolevulinic acid cream, long-term cure rates were reported to be up to 89 percent using blue or red light. Treatments for 10–14 days provided excellent cosmetic results with all lesions developing erythema and crusting 2 to 4 days after treatment [47]. Many individuals did experience a stinging and burning pain during and after irradiation, but this effect was generally below accepted thresholds [48].

### 2.9. Skin Rejuvenation

Light treatment has been used to improve the appearance of skin especially of individuals who have photoaged skin [44]. LEDs have an advantage in that they do not produce thermal injury. The skin-rejuvenating effects of LED systems are produced by photobiomodulation [49]. The nonthermal process of photobiomodulation results in the excitation of endogenous chromophores which produce photophysical and photochemical events. Photobiomodulation also stimulates processes in the skin such as fibroblast proliferation, collagen synthesis, growth factors, and extracellular matrix production [50]. It produces these effects by activating cellular mitochondrial respiratory pathways which results in skin tightening. In a small trial, participants were irradiated with the 830 nm LED on days 1, 3, 5, 15, 22, and 29 and with 633 nm LED days 8, 10, and 12. Both treatments were for a 20 min period. Most participants showed an effect on periorbital wrinkles, with improvement in photo aging and softness scores. Mild redness in skin colour, was reported by some participants [51].

## 3. New Phototherapy Devices for Dermatological Conditions

### 3.1. Phototherapy Devices for Treatment of Allergenic Rhinitis

An example of new phototherapy devices is shown on Figure 1 (Allergy Reliever). This is a Class IIA medical device (Kodec Holdings, Unit 01, 14/F., Tai Ping Industrial Centre, Block 1, No 57 Ting Kok Road, Tai Po, New Territories, Hong Kong). This phototherapy device has two specific wavelengths which are recommended for reducing the symptoms of allergic rhinitis. The device can be used directly on to the skin in the nasal cavity for a 3 min period. The device emits visible light (mUV/VIS) and infrared light (660 nm and 940 nm) wavelengths. It is recommended that it’s used twice a day, 5 to 6 h apart. The treatment was performed twice a day with 5–6 h between each treatment [52,53]. This device was shown to be effective and can be used directly by the sufferer as an additional treatment for allergenic rhinitis however with an appropriate design emitting the correct wavelength it could be used for treatment of wounds which represent a significant burden to patients, health care professionals, and health care systems. For example, the device could also be adapted for use on diabetic patients with foot ulcers. Foot ulcers are a disabling complication of diabetes that affect 15% to 25% of people with diabetes at some time in their lives. There is some evidence that ultraviolet light, and blue light can help ulcers heal through multiple mechanisms such as increased cell growth and blood vessel activity [54].

There was a good relationship between the symptoms reported by the participants in their allergy histories and symptoms provoked in an Allergen Challenge Chamber during a baseline visit. Sixty-four data sets were collected. Total nasal symptom scores (TNSS) were collected at the end of the treatment period. The TNSS scores showed that there was little or no change in the intensity of symptoms scored at the baseline and at the final study visit for participants in the placebo group who used a placebo device, which did not emit light. The difference in the intensity of symptoms scored at the baseline and at the final visit for the group, using the photoperiod device was significantly lower (Table 1) with a downward trend in the intensity of symptoms [52]. This demonstrates that these types of devices can be effective in allergenic rhinitis treatment regimes.

### 3.2. Other New Phototherapy Devices

Several new devices have become commercially available. Most of these devices have published efficacy and safety data. It is important that LEDs are safe to use. LEDs used alone do not cause damage to the epidermis or dermal tissues and have no adverse events associated with their use [1]. When LED phototherapy is used patients can be treated quickly and this can fit into their working day.

#### 3.2.1. BF-RhodoLED^®^ (Red Light PDT Illuminator)

BF-RhodoLED has a narrow emission spectrum closely centred around 635 nm ± 9 nm which is optimal for the stimulation of living cells. The BF-RhodoLED is designed for the delivery of 37 J cm^−2^ required for ALA PDT with narrow-spectrum lamps for 10 min when using the standard treatment protocol. This is a standard approach in delivery of phototherapy and is designed to promote safe application which could be used as a “do it yourself treatment” [55]. Multiple actinic keratosis showed complete clearance rate after a maximum of two PDTs using the BF-RhodoLED^®^. Cosmetic outcomes were improved in the PDT group compared with placebo although treatment-emergent adverse events around the treatment site were experienced in the clinical study. These were usually minor irritations but there should be further investigation of these effects [55].

#### 3.2.2. Aktilite^®^ (Fractional Carbon-Dioxide (CO_2_) Laser)

Recent treatments for Bowen’s disease include fractional carbon-dioxide (CO2) laser treatment with PDT to increase treatment efficacy by making penetration channels through the epidermis. Prior to application of the photosensitizer, the epidermis of the lesions was totally removed by using a fractional CO_2_ laser with a tip size of 120 µm and a peak power of 30 W which produced pulsed energy of 50 mJ. Methyl aminolevulinate was applied for 90 min followed by light exposure. There was no significant difference between using Methyl Aminolevulinate compared to 5- aminolaevulinic [53]. A 630 nm light was utilized at a light dose of 37 J/cm^2^. Each treatment had a gap of four weeks. Result produced 87.5% with a complete response [56].

#### 3.2.3. Blue U^®^ (Pulsed Light and Blue Light Activation)

Using blue light after intense pulsed light in an initial treatment in photodynamic therapy with short-contact 5-aminolevulinic acid in 3 subsequent treatments resulted in significant improvement in severity of acne. There was a reduction in the number of lesions and improvement in skin texture with smoothing of scar edges where severe (class 4) facial cystic acne and scarring had occurred [57].

## 4. Discussion

Phototherapy is a useful method in the management of dermatological conditions in both adults and children and has gained widespread acceptance for some conditions. Studies demonstrate that NB-UVB is effective in the treatment of *Psoriasis vulgaris*, vitiligo, lichen planus, chronic eczema and annular granuloma, while PUVA phototherapy is useful for localized scleroderma, *Psoriasis vulgaris*, vitiligo, cutaneous T cell lymphoma and acute atopic dermatitis [50]. Recently it has been shown that blue light treatment regulates proliferation and differentiation in human skin cells [57]. There is a paucity of information on how children and adults differ in their sensitivity to different wavelengths and the optimal dosages for different conditions [8,9,10]. Lesions caused by acne and some proliferative skin diseases improve when exposing only specific parts of the skin to blue light under controlled conditions [18]. Clinical trials have shown the use of blue light for *Helicobacter pylori* stomach infections had beneficial effects [58]. Blue light inactivation of the important wound pathogenic bacteria *Staphylococcus aureus* and *Pseudomonas aeruginosa* have also been reported [59]. It has also been shown to be useful in wound healing through the regulation of epidermisation and its effects on keratinocytes as these cells are responsible for restoring the epidermis. Blue light may also cause other phenotypic changes. However, blue light has also been reported to cause cell dysfunction by the photoexcitation of blue light sensitizing chromophores, including flavins and cytochromes, within mitochondria or/and peroxisomes [60]. Other damaging effects of phototherapy have also been reported [61,62]. The introduction of light-emitting diode (LED) devices has reduced many of the concerns formerly associated with standard light therapies, such as expense, safety and the need for trained personnel to operate them. Many LED devices are designed for home use and are widely sold on the internet. It is possible that combined wavelengths could be used and some of these could be useful in treatments that are self-administered such as the allergy reliever device (Figure 1). The allergy reliever device is a useful treatment for those sufferers who cannot take other medication. Medicines such as steroids and antihistamines are traditionally prescribed as over the counter medical therapies but there many sufferers who do not wish to take medication or for who medication is contraindicated. There are also allergic rhinitis sufferers who wish to reduce the amount of medication that they take, or who find that medication is not sufficient to control their symptoms [52]. However, care has to be taken in self-administering dosages of light as skin types vary greatly. Young skin may react in different ways to specific wavelengths which can interact to produce other forms of damage to the skin. Wavelengths beyond the UV spectrum can damage human skin. These include the blue light region of visible light (VIS) as well as the near infrared range (IRA). The effects of prolonged periods of exposure to combinations of wavelengths is not clear and requires further investigation. However, it has been shown that IR-A radiation has beneficial effects on collagen metabolism. Studies suggest children may be treated safely and effectively with various forms of phototherapy for common skin conditions, including psoriasis, atopic dermatitis, and vitiligo [63]. Phototherapy can reduce disease burden in children and adults with severe atopic dermatitis and psoriasis and should be considered as a second-line therapy if standard regimens are unsuccessful [64,65]. However, the longer-term risks from phototherapy treatments in younger children need further research before treatment could be widely recommended.

## 5. Conclusions

Phototherapy has a broad range of usages in medical and other skin conditions. The efficacy of phototherapy depends on the combination of wavelength, frequency, irradiation time and the dose. Three light wavelengths have demonstrated several therapeutic applications. These are blue (415 nm), red (633 nm), and near-infrared (830 nm). The full action of visible light is not understood in comparison to UVB. UV-free blue light phototherapy (wavelength 400–500 nm) has also been shown to have a wide range of use as an effective skin treatment. Phototherapy is generally safe but offers the possibility of being directly administered by the sufferer provided that certain safety aspects are accounted for within the treatment device. Further evaluation of the safety aspects would require additional research particularly if treatments were applied to children. Phototherapy can be used to rejuvenate aged skin especially when used in conjunction with a photosensitizer, such as 5-ALA.

## Figures and Tables

**Figure 1 life-13-00196-f001:**
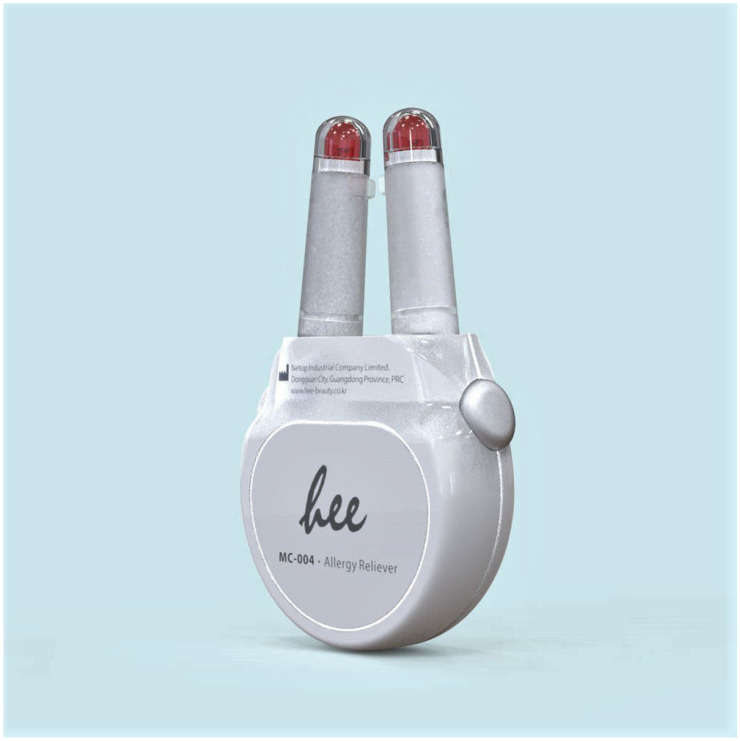
Phototherapy device (Allergy Reliever).

**Table 1 life-13-00196-t001:** Mean, Total nasal symptom scores (TNSS) and *p* values for placebo and treatment groups.

Placebo Group					Treatment Group			
Allergen Type	Mean Score (Baseline)	Mean Score (Final Visit)	Total TNSS Scores at (Baseline)	Total TNSS Score (Final Visit)	Mean Score (Baseline)	Mean Score (Final Visit)	Total TNSS Scores (Baseline)	Total TNSS Score (Final Visit)
Grass only	7	6	57	46	7	4	40	21
Grass and cat/house dust mite	7	7	123	120	8	5	144	99
Cat/house dust mite only	8	7	58	43	8	4	56	22
Allergen	Significant difference at baseline between placebo group and treatment group *p*-value				Significant difference at final visit between placebo group and treatment group *p*-value			
Grass only	0.603				0.138			
Grass and cat/house dust mite	0.312				0.009			
Cat/house dust mite only	0.624				0.144			

## Data Availability

Not applicable.

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
