# Peer review of "Phototherapy as a Treatment for Dermatological Diseases, Cancer, Aesthetic Dermatologic Conditions and Allergenic Rhinitis in Adult and Paediatric Medicine"

_life, 2023, doi:10.3390/life13010196_

Round 1
Reviewer 1 Report
The paper aims to investigate the role of phototherapy in dermatological diseases and allergic rhinitis, both in adult patients and children, but several aspects need attention and improvement. My comments are the following: Introduction paragraph: it is confusing, with a continuous lack of references. Initially, the author states that UVB for the treatment of psoriasis is effective and later states that there is only some efficacy. “Narrow Band ultraviolet B (NBUVB) (311–313 nm) is the most studied and is frequently prescribed as an effective treatment for pediatric patients with psoriasis”…“However there is no consensus on the use of phototherapy on pediatric psoriasis but NBUVB, BBUVB, and PUVA 75 have been shown to have some effect in the treatment of pediatric psoriasis”. Please explain it better with adequate references.
Line 87: “Phototherapy is effective after maximizing topical therapies and can be administered when needed”. Please explain the meaning better. Paragraph 2: it is characterized by a series of pathologies, which are neither described nor, above all, the mechanisms by which phototherapy leads to disease and symptoms are described. Lichenoid pityriasis paragraph: does the entire paragraph refer to a single reference? Please insert the reference at the end of each sentence. Line 114: the abbreviation NB-UVB had already been used previously in the text. Please correct and check all abbreviations. Line 119: Please insert the appropriate reference at the end of the sentence. Use of Near Infra-Red (NIR) wavelengths in phototherapy treatments paragraph: the author writes about bruxism. What is the link to the rest of the text? Line 137-139: A double-blind randomized study about the role of low-energy narrow-band phototherapy at 660 nm has been inserted and repeated in lines 192-195. Please correct. Line 152: Please insert the appropriate reference at the end of the sentence. Lines 201-203: Please insert the appropriate reference at the end of the sentence. Line 213: Please rewrite the sentence. Photodynamic therapy paragraph: The chapter is incomplete and unclear. Furthermore, since the paper aims to treat the skin pathologies of child-age as well, all the pathologies of the child treated with PDT, for example viral warts, should be added. Line 213: Please rewrite, the verb is missing. Line 219-221: Please add the appropriate reference. Phototherapy devices for treatment of allergenic rhinitis paragraph: the author wrote that ultraviolet light and blue light can help ulcers heal through multiple mechanisms such as increased cell growthand blood vessel activity, but he didn’t explain the mechanisms through which this happens. Table 1: the image quality needs to be improved significantly, as it is evident that it is a screenshot. Other New Phototherapy Devices paragraph: “LEDs used 299 alone do not cause damage to the epidermis or dermal tissues and have no adverse events 300 associated with their use”. Please insert the reference to support this claim. BF-RhodoLED®, Aktilite®, Blue U® paragraphs: all of them should be better investigated, as they may represent the most innovative part of the manuscript. Discussion paragraph: Line 341: Please add the appropriate reference. Line 345: Please add the appropriate reference. Line 347: Please add the appropriate reference. Line 348-350: which are the damaging effects reported? A conclusion paragraph is missing. Please add. The title of the paper fits little with what the text is about. Moreover, the pediatric aspect is investigated only in phototherapy, not PDT or new therapies. Furthermore, the aesthetic aspect is little taken into consideration throughout the test.
Author Response
See File attached

Reviewer 2 Report
Phototherapy as a treatment for aesthetic dermatologic conditions and allergenic rhinitis in adult and paediatric medicine
I suggest minor revisions:
· This review is interesting and contributes to the knowledge of phototherapy. In that sense, I think the title should be wider. The manuscript not only includes Phototerapy as a treatment for aesthetic dermatologic conditions and allergenic rhinitis, but the author also mentions dermatologic diseases and cancer, for example. I suggest changing the title.
· Page 8, line 321: The author mentions methyl aminolevulinato applied in PDT, and previously describes PDT with 5-aminolevulinic acid, please clarify the difference between these compounds and their use.
· Some paragraphs lack references:
Page 1, lines 36-38: The three wavelengths of light that have demonstrated several therapeutic applications are blue (415nm), red (633nm), and near-infrared (830nm).
Page 2, lines 43-44: Several reviews have published evidence on the effectiveness of PBM treatments in mitigating inflammation and promoting wound healing.
Page 3, lines 114-117: Both PUVA and narrow band UVB (NBUVB) have been shown to be as effective treatments for stage I paediatric cutaneous T-cell lymphoma (CTCL). Narrow Band ultra violet B is easier to administer and for this reason can be considered as the preferred treatment.
Page 3, lines 117-119: NBUVB phototherapy, combining ultraviolet A1 (UVA1) and UVB, 308-nm phototherapy and topical PUVA have all been used effectively in the treatment of childhood vitiligo (loss of skin pigment cells) and pruritus (unpleasant skin sensation).
Page 3, lines 119-122: Phototherapy has also been used successfully to treat localized scleroderma, morphea, nodular prurigo, Langerhans cell histiocytosis, and cutaneous graft-versus-host disease (GVHD) in paediatric patients.
Page 3-4, lines 149-152: In-vitro studies using infra-/red light have not shown any useful impact on wound healing through an effect on inactivated cells located in deeper skin layers like fibroblasts or stem cells.
Page 4, lines 156-159: Clinical trials using blue light indicate promising results for the development of a system for improved wound healing through light stimulation. Blue light can be used to prevent non optimal epidermisation in premature healing stages by slowing down the cell metabolism.
Page 5, lines 234-235: Many individuals did experience a stinging and burning pain during and after irradiation but this effect was generally below accepted thresholds.
Page 7, lines 298-301: Several new devices have become commercially available. Most of these devices have published efficacy and safety data. It is important that LED’s are safe to use. LEDs used alone do not cause damage to the epidermis or dermal tissues and have no adverse events associated with their use.
Page 8, lines 341-342: Lesions caused by acne and some proliferative skin diseases improve when exposing only specific parts of the skin to blue light under controlled conditions.
Page 8, lines 343-344: Clinical trials have shown the use of blue light for Helicobacter pylori stomach infections had beneficial effects.
Page 8, lines 344-345: Blue light inactivation of the important wound pathogenic bacteria Staphylococcus aureus and Pseudomonas aeruginosa have also been reported.
Author Response
See file attached
